# Humanizing the yeast origin recognition complex

Clare S. K. Lee [1,7], Ming Fung Cheung [2,3,7], Jinsen Li[4], Yongqian Zhao[3,5], Wai Hei Lam[1], Vincy Ho[2,3], Remo Rohs [4], Yuanliang Zhai [1✉], Danny Leung [2,3✉] & Bik-Kwoon Tye [3,5,6✉]

The Origin Recognition Complex (ORC) is an evolutionarily conserved six-subunit protein complex that binds specific sites at many locations to coordinately replicate the entire eukaryote genome. Though highly conserved in structure, ORC's selectivity for replication origins has diverged tremendously between yeasts and humans to adapt to vastly different life cycles. In this work, we demonstrate that the selectivity determinant of ORC for DNA binding lies in a 19-amino acid insertion helix in the Orc4 subunit, which is present in yeast but absent in human. Removal of this motif from Orc4 transforms the yeast ORC, which selects origins based on base-specific binding at defined locations, into one whose selectivity is dictated by chromatin landscape and afforded with plasticity, as reported for human. Notably, the altered yeast ORC has acquired an affinity for regions near transcriptional start sites (TSSs), which the human ORC also favors.

[1] School of Biological Sciences, The University of Hong Kong, Pok Fu Lam Road, Hong Kong. [2] Center for Epigenomics Research, The Hong Kong University of Science & Technology, Clear Water Bay, Hong Kong. [3] Division of Life Science, The Hong Kong University of Science & Technology, Clear Water Bay, Hong Kong. [4] Quantitative and Computational Biology, Departments of Biological Sciences, Chemistry, Physics & Astronomy, and Computer Science, University of Southern California, Los Angeles, CA 90089, USA. [5] Institute for Advanced Study, The Hong Kong University of Science & Technology, Clear Water Bay, Hong Kong. [6] Department of Molecular Biology & Genetics, Cornell University, Ithaca, NY 14853, USA. [7]These authors contributed equally: Clare S. K. Lee, Ming Fung Cheung. ✉email: zhai@hku.hk; dcyleung@ust.hk; biktye@ust.hk

The regulation of the location and timing of DNA replication initiation is critically important for the growth and development of eukaryotes. DNA replication initiation is regulated stringently, involving multiple layers of redundant mechanisms[1]. During development, metazoans utilize different sets of replication origins based on accessible chromatin that is configured developmentally in stages[2,3]. In contrast, the budding yeast *Saccharomyces cerevisiae*, which does not have a distinct developmental program, initiates DNA replication at a defined set of locations known as autonomously replication sequences (ARSs)[4]. How metazoan cells attain plasticity in origin selection in response to programmed development is unknown.

The origin recognition complex (ORC), which binds replication origins in eukaryotes consists of six highly conserved subunits, Orc1, 2, 3, 4, 5, and 6 (ref. [5]). Despite their conservation in sequence and in structure[6–8], ORC has divergent binding specificities that vary from species to species[9]. With the exception of *Schizosaccharomyces pombe*, which has multiple AT-hooks in Orc4 (ref. [10]), in most model yeast species, ORC binds defined and species-specific base sequences[11,12]. In *S. cerevisiae*, ARS is characterized by a 17 bp ARS consensus sequence (ACS) embedded within ~125 bp DNA with a polar T-rich and A-rich base composition that is asymmetrically located between two well-positioned nucleosomes[13]. In metazoans, ORC binds selectively and stochastically to open chromatin regions containing active chromatin marks, but not specific nucleotide sequences[14–17]. Other features that associate with ORC in human cells include CpG islands, transcription start sites (TSSs) and GC-rich DNA, all tending to be in nucleosome-depleted regions (NDRs)[3,16,18,19]. In contrast, there is a paucity of ORC-binding sites in genomic regions that associate with common fragile sites and recurrent cancer deletions[16,19]. In human, the bromo adjacent homology domain of Orc1 is believed to play a role in tethering ORC to H4K20me2 chromatin[20], but the molecular mechanism that targets ORC to specific loci is poorly understood.

Important information that could explain the disparate binding specificities between yeasts and metazoans is provided by the atomic model for the *S. cerevisiae* ORC bound to ARS305 that detailed the interactions between the subunits of ORC and the ACS[8]. In this model, ORC binds the ACS through multiple nonspecific and specific interactions. The multiple nonspecific interactions between Orc1-5 with the sugar phosphate backbone of the ACS provide a fairly firm grip without base specificity. The base-specific contacts are carried out by the basic patch 4 (BP4) of Orc1 and the initiator-specific motif (ISM) of Orc2 in the minor groove, and the insertion helix (IH) of Orc4 in the major groove. Importantly, these interactions with the ACS and the interactions of Orc2, Orc5, and Orc6 with the B1 element bend the ARS DNA at the ACS and B1. This DNA bending by ORC is also observed in metazoans[21], suggesting that it may be obligatory for the loading of the MCM double hexamer. Consistent with this hypothesis, the loaded DNA in the ORC-Cdc6-Cdt1-MCM structure after capture by the MCM ring is straight[22].

Comparison of sequence alignments of these DNA-contacting motifs among eukaryotes showed that only the Orc1-BP4 is conserved, while the Orc2-ISM and Orc4-IH have diverged[8]. In particular, the Orc4-IH, encoded by 19 amino acids, which interacts with methyl groups of the invariant thymines of the ACS, has diverged in sequence among yeasts and absent in *S. pombe* and metazoans (Fig. 1a, b). Considering that A–T base pairs in the minor groove have indistinguishable hydrogen acceptors between A–T or T–A base pairs, the only relevant motif for base recognition would be the Orc4-IH in the major groove of the ACS. Comparing the winged helix domains (WHDs) of ScOrc4, DmOrc4, and HsOrc4, we found that they are highly conserved in structure except for the extra IH in ScOrc4-WHD

(Fig. 1a). Removal of the IH would likely confer metazoan/human-like binding properties to the yeast ORC.

## Results

### Orc4-IHΔ strain is viable with activated S-phase checkpoint.
To test the hypothesis that yeast ORC can be humanized by removal of the Orc4-IH, we constructed yeast strains with the endogenous *ORC4* under the regulation of the *GALS* promoter (*GALS-ORC4*), and expressing the mutant Orc4-IHΔ or wild-type (WT) Orc4 ectopically under the native *ORC4* promoter. The *GALS-ORC4* strain with empty vector was made as a negative control (Fig. 2a). Previous study showed that point mutations in BP4 of Orc1 is lethal[23]. Hence, we also constructed Orc1-BP4Δ expressing strain with endogenous *ORC1* under regulation of *GALS* promoter as a negative control (Fig. 2b). In contrast to Orc1-BP4Δ mutant, which is inviable when *GALS* promoter is switched off, the Orc4-IHΔ mutant grows slower with normal S-phase progression, but a longer G2/M phase than the WT control (Fig. 2c and Supplementary Fig. 1b). To investigate if the prolonged G2/M phase phenotype is the result of intra-S-checkpoint activation, we examined the phosphorylation state of Rad53 (ref. [24]). At 110 min after release from G1, Rad53 is clearly phosphorylated in the mutant but not the WT control (Fig. 2d), suggesting that DNA damage accumulated at the end of S-phase activated the S-phase checkpoint, consistent with the G2/M delay in the mutant. The mutant also shows slight hydroxyurea (HU) sensitivity as might be expected with an activated S-phase checkpoint (Fig. 2e).

### ORC-IHΔ binds promiscuously with a preference for TSSs.
To determine if the IH deletion could affect the ability of ORC in chromatin binding, we first examined the chromatin retention of ORC in both WT and mutant strains. The mutant ORC appears to behave similarly as its WT counterpart in DNA binding based on its retention on chromatin (Supplementary Fig. 2a). Next, we carried out chromatin immunoprecipitation-sequencing (ChIP-seq) of ORC in G2/M phase from both the WT and mutant strains to elucidate the ORC genome-wide binding profiles (Fig. 3a and Supplementary Fig. 3). In total, we defined 618 and 2219 ChIP-seq peaks in WT and mutant, respectively (Fig. 3b). Compared to previously published datasets, WT ORC ChIP-seq shows high degree of consistency and affinity toward binding to a known set of ACSs defined by stringent criteria, including T strand polarity and precise nucleosome positioning[13] (Fig. 3a, c and Supplementary Fig. 3a–c). However, the mutant ORC-binding profile is dramatically different (Fig. 3a, b and Supplementary Fig. 3d, e), with significantly reduced enrichment at ACS sites (Fig. 3c). Furthermore, we discovered many novel ORC peaks located in broad clusters (Fig. 3a) within intergenic regions (Supplementary Fig. 4a, b). Human studies showed that ORC-binding sites are enriched near TSS[16,19], a feature not observed for *S. cerevisiae*[13]. However, we found that relative to WT ORC, ORC-IHΔ binds with distinct enrichment both upstream and downstream of transcripts that positively correlate with the transcriptional strength of the associated genes (Fig. 3d). In fact, 1839 out of 2219 (83%) of the mutant ORC-binding sites are located within 500 bp upstream of a TSS. The high enrichment of ORC-IHΔ upstream of TSSs is especially interesting because TSSs are also known to have a similar polar bias in T-rich and A-rich sequence, and well-positioned downstream nucleosomes as origin DNA in *S. cerevisiae*[13]. This result suggests that the mutant ORC has lost its sequence specificity and binds most highly to open chromatin that may harbor certain properties shared with origin DNA. The large number of highly distributed ORC-IHΔ-binding sites compared to the limited ORC molecules per cell[25] suggests

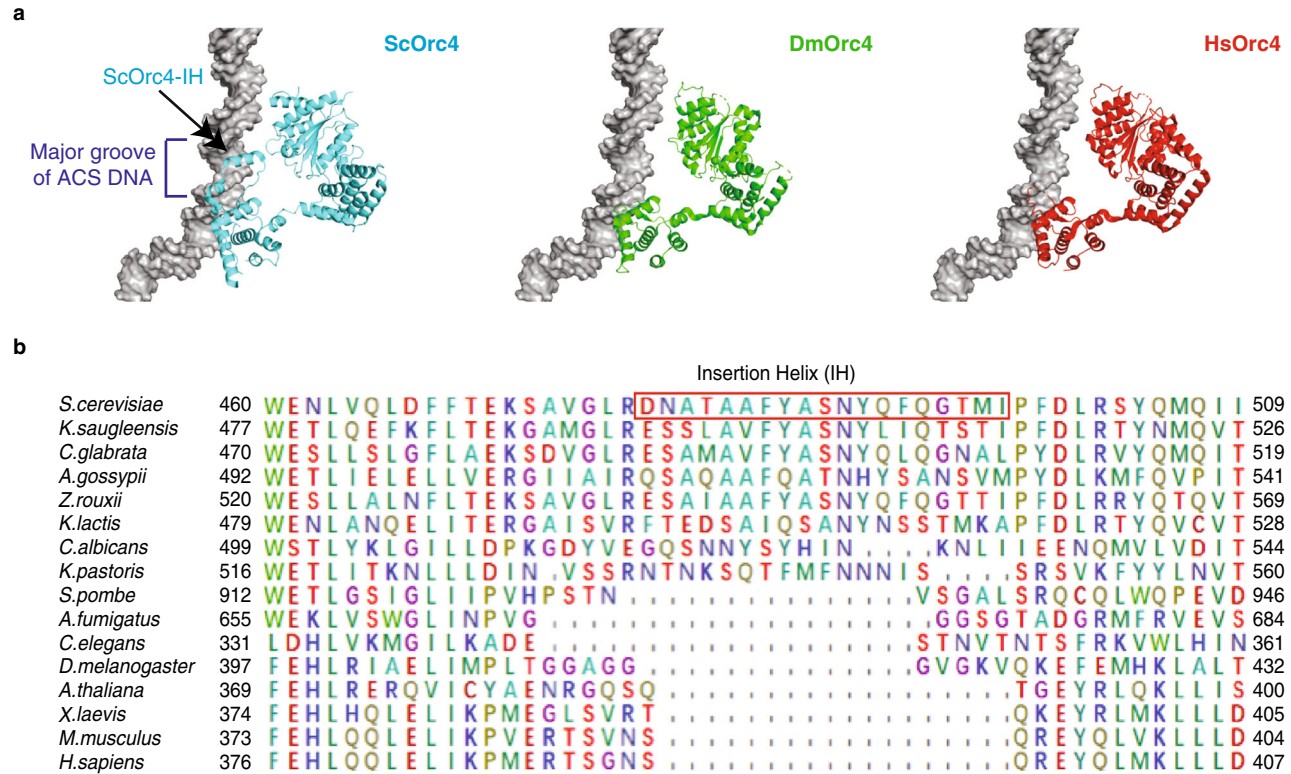

**Fig. 1 Orc4-IH is divergent in yeasts and absent in metazoans. a** WHDs of Orc4 from *S. cerevisiae* (Sc), *Drosophila melanogaster* (Dm), and *Homo sapiens* (Hs) in contact with DNA. **b** Multiple protein sequence alignment of Orc4 insertion helixes from indicated species.

that ORC-IHΔ binding in the mutant strain occurs promiscuously and stochastically.

**ORC-IHΔ has a propensity for T-rich sequences**. To characterize the DNA sequences bound by the mutant ORC, we performed in depth analyses on ORC-binding sites defined in both WT and mutant. Motif enrichment analysis identifies the canonical ACS (TTTATGTTTAG) in WT ORC peaks. No known motifs are detected in the ORC-IHΔ peaks, with only a propensity for poly-T tracts and GC-rich sequences, such as (TTTTTYSS) (Fig. 4a). We identified 183 sites with ACS-like motif (TTTATGTTAGK) in WT ORC peaks and 2555 sites with the novel (TTTTTYSS) motif among the ORC-IHΔ peaks. Standardizing the motif's strand orientation, we investigated the surrounding regions' base composition with the position 0 denoting the first base of the motif. As previously reported[26], we found an A-rich region downstream of the WT motif. The same T- to A-rich bias is not as obvious in the mutant although the T-richness is as prominent if not more (Fig. 4b and Supplementary Fig. 5). Based on the ORC-DNA atomic model, the T-richness of the mutant ORC-binding motif could potentially support the interaction of the conserved and essential Orc1-BP4 (Fig. 2b) with the minor groove of origin DNA to provide a degree of stability for the mutant ORC without the 19aa IH of Orc4. To biochemically verify these predicted ORC-binding sequences, we carried out electrophoretic mobility shift assays (EMSA) with two 30 bp probes containing either the canonical ACS (TTTATATTTAG) of ARS607 or a novel sequence (TTTTTTTTCCGCG) identified in the ORC and ORC-IHΔ ChIP-seq peaks, respectively (Supplementary Fig. 6). We find that the ORC-IHΔ binds to the ACS with affinity comparable to the WT (Supplementary Fig. 6a, b). However, it binds with a higher affinity to the TTTTTTTTCCGCG sequence, to which the WT ORC shows reduced affinity. Therefore, while retaining the capacity to bind ACS DNA, these

results suggest an acquired affinity of the mutant ORC, which may be associated with the shape rather than the nucleotide sequence of the DNA. It is important to emphasize that in vitro binding assays do not accurately measure the selectivity of ORC in vivo, given that the protein concentrations are vastly different in these two systems. Moreover, the accessibility and structures differ between naked DNA and chromatin. Nonetheless, a closer look at the shapes of the two substrates can be informative. Predictions using Monte Carlo simulations[27–29] show that the DNA shape preferred by ORC-IHΔ has narrower minor groove width (MGW), more negative propeller twists (ProTs) and roll angles, increased helix twist (HelT) and enhanced negative electrostatic potential, as compared to ACSs (ARS607, ARS305, and ARS416; Fig. 4c). These composite geometric and biophysical properties may potentially contribute to a distinct shape that deviates from the canonical ACS and is recognized by the ORC-IHΔ. The differential binding substrates of WT and mutant ORC are presented in a modeled illustration (Supplementary Fig. 7).

**ORC-IHΔ is proficient in MCM loading**. A central function of ORC is to serve as a platform for the loading of the MCM double hexamer, a process known as pre-replicative complex (pre-RC) assembly. To determine if ORC-IHΔ is capable of promoting efficient MCM loading, we examined the chromatin loading in nuclear extracts and pre-RC assembly on linear DNA, using purified proteins. Similar levels of MCM complexes associating with G1 chromatin are observed in both WT and mutant strains (Supplementary Fig. 2a). Consistently, both the WT and mutant ORC exhibit comparable activities in Cdc6-dependent loading of MCM onto DNA (Supplementary Fig. 2b) and assembly of the MCM double hexamer (Supplementary Fig. 2c). These results indicate that the function of ORC in pre-RC assembly is not substantially affected by the IH deletion although the efficiency of

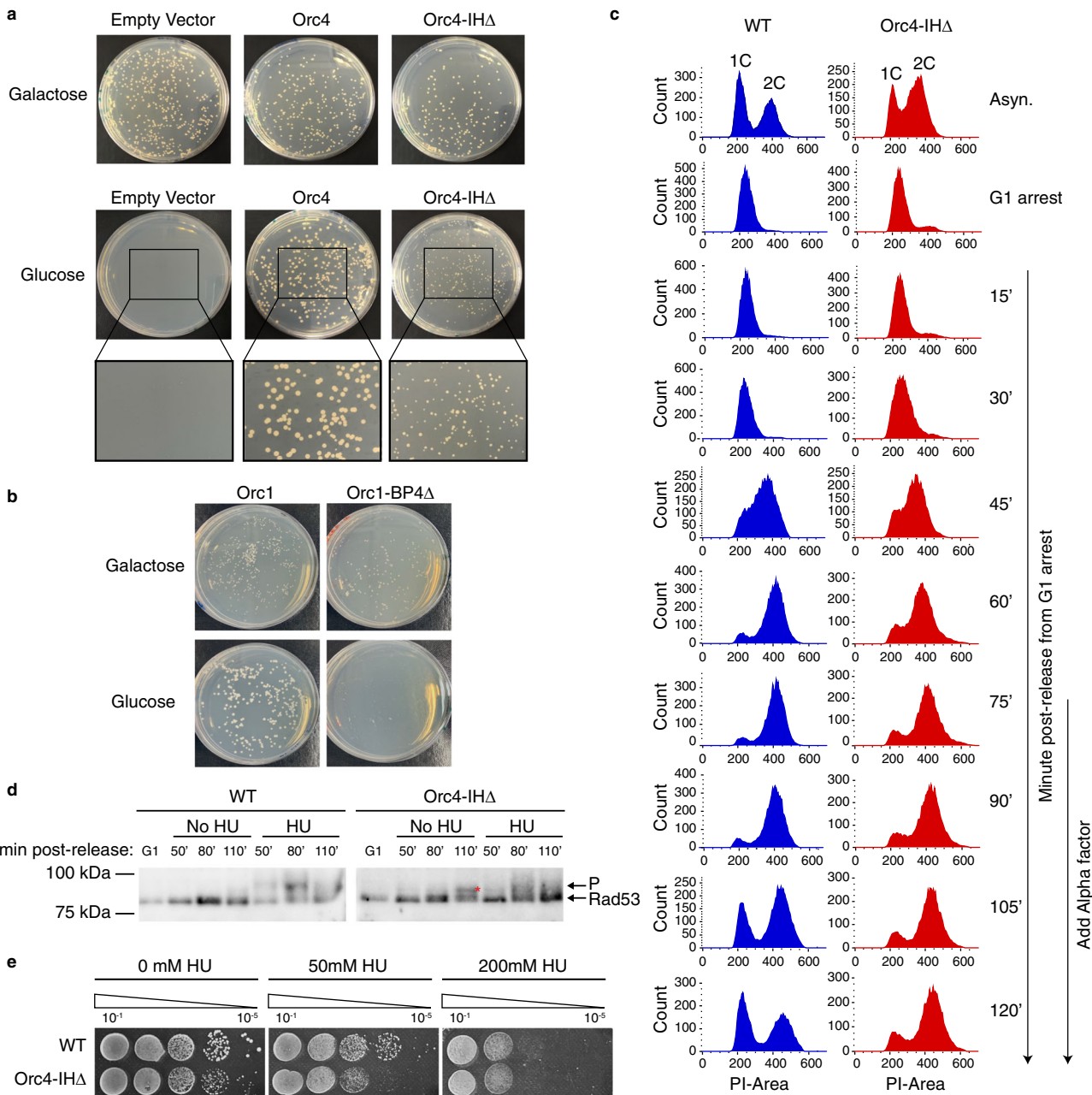

**Fig. 2 Yeast strain with Orc4-IHΔ has normal S-phase but prolonged G2 phase. a** GALS-ORC4 yeast strains ectopically expressing Orc4, Orc4-IHΔ, or vector alone control were plated on SCM-Ura plates containing either galactose or glucose. Zoom-in views of 2× magnification were shown for the glucose selection plates. **b** GALS-ORC1 yeast strains ectopically expressing Orc1 or Orc1-BP4Δ were plated on SCM-Ura plates containing either galactose or glucose. **c** FACS analysis shows Orc4-IHΔ cells exhibit normal S-phase progression yet delayed G1 entry from G2M. WT and mutant cells were first arrested in G1 and then released into S-phase. Alpha factor was added to the cultures at 75 min post release from G1 to monitor G1 entry from G2M. **d** S-phase checkpoint activation assayed by Rad53 phosphorylation. WT and mutant cells were arrested in G1 using alpha factor (G1), and released into YPD without hydroxyurea (no HU) or YPD containing 200 mM HU (HU) for the indicated times at 30 °C. Trichloroacetic acid (TCA) lysates were prepared for western blotting. Rad53 is phosphorylated in mutant, but not in the WT in the absence of HU. (*) denoted the phosphorylated state of Rad53. Blots shown are representative of n = 3 biological replicates. Source data are provided as Source data file. **e** HU sensitivity is assayed by spotting serially diluted WT and mutant cells onto YPD plates containing the indicated concentrations of HU.

pre-RC assembly could be quite different in vivo because of the difference in protein concentrations and template accessibility.

**MCM loading and replication initiation are limited and stochastic.** To achieve a genome-wide perspective of the MCM loading in the Orc4-IHΔ strain, we performed MCM ChIP-seq in both WT and mutant cells arrested in G1 phase (Fig. 5 and Supplementary Fig. 2d). Integrative analysis shows that higher

ORC and MCM enrichment is found at ARS regions in WT, supporting the notion that ORC binding in G2/M promotes MCM loading in G1 (Fig. 5a). This dependence of MCM recruitment on ORC binding is best illustrated by focusing on the ORC peaks that are unique to the WT (Fig. 5bi). While concordant ORC and MCM enrichment is detected at these sites in WT, we observed little to no binding of either complex in the mutant strain (Fig. 5bi). At ORC peaks that were overlapping, but

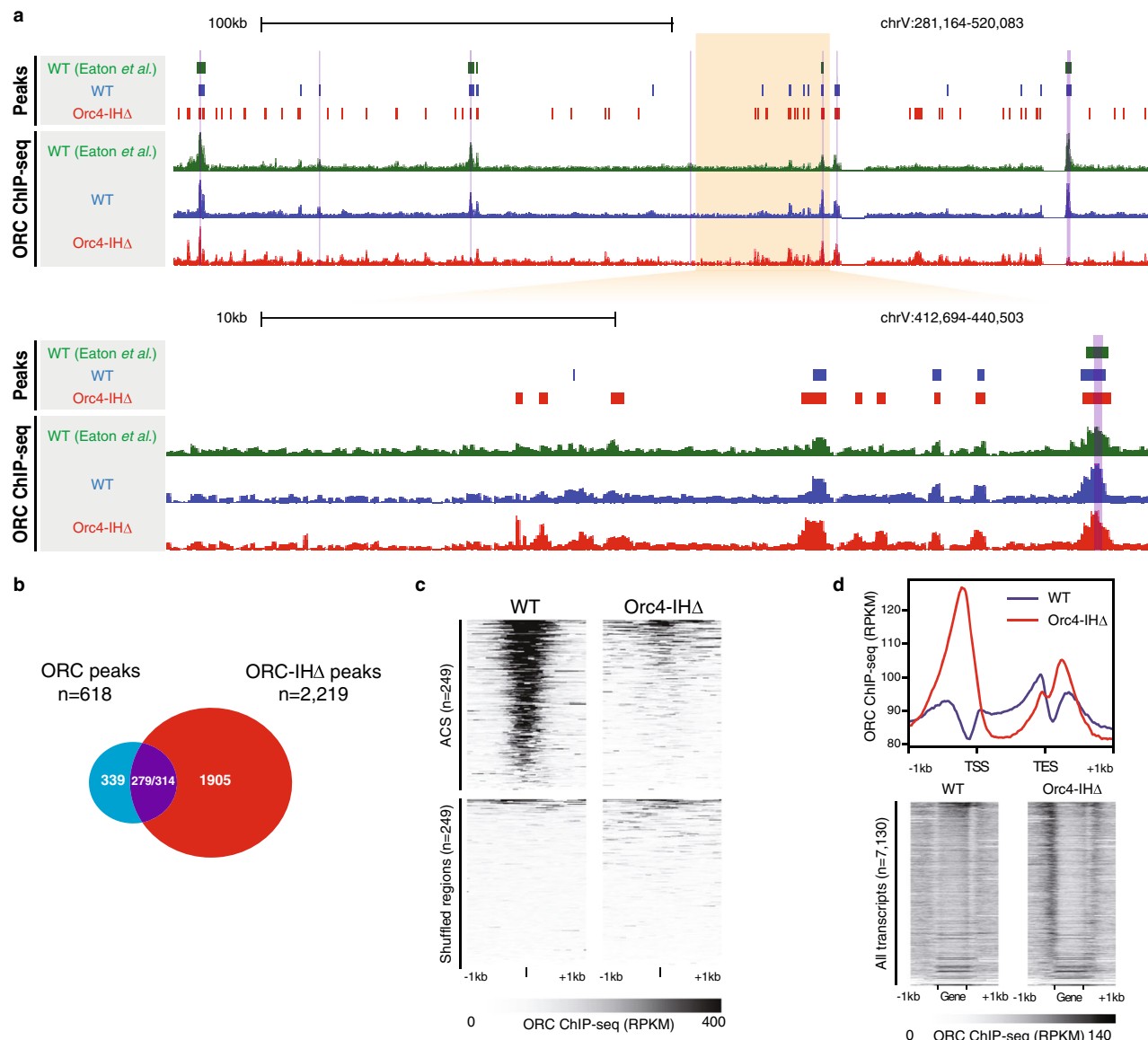

**Fig. 3 Orc4-IH motif deletion alters the DNA-binding specificity of ORC. a** Genome browser screenshots illustrate the enrichment patterns of ORC across the yeast genome. Normalized signals (Reads per kilobase per million reads (RPKM)) of WT ORC (blue), ORC-IHΔ (red), and previously published ORC ChIP-seq datasets (green)[13] are shown alongside their corresponding peak calls. Y-axis for all ChIP-seq tracks within the screenshot are set to the same scale. **b** Venn diagram showing the overlap between ORC ChIP-seq peaks defined in WT (n = 618; blue) and Orc4-IHΔ cells (n = 2219; red). Of these, 279 ORC and 314 ORC-IHΔ peaks overlapped by at least 1 bp (purple), while 339 and 1905 peaks were identified as unique in WT and Orc4-IHΔ, respectively. **c** Heatmaps show the ORC enrichment patterns in WT and Orc4-IHΔ at regions containing previously defined ACS (±1 kb of ACS; top). Size- and number-matched shuffled genomic loci were included as controls (bottom). **d** Line plot and heatmaps show the ORC enrichment patterns in WT and Orc4-IHΔ at annotated transcripts and flanking sequences. The line plot (top) shows the aggregated ORC ChIP-seq signal (RPKM) at all annotated transcripts and the surrounding regions (±1 kb). Orc4-IHΔ binding (red) at the 5′ of the transcriptional start site (TSS) is dramatically higher than that of WT (blue). Heatmaps (bottom) also demonstrate the differential enrichment of Orc4-IHΔ upstream of the TSS. The transcripts in the heatmaps are arranged by descending RNA-seq signals, indicating the association of mutant ORC binding at these loci and the expression levels.

may or may not be overlapping in binding sites in WT and mutant, there is also a strong concordant ORC and MCM binding in both strains (Fig. 5bii). As expected, ORC peaks that are unique to Orc4-IHΔ show weak to no ORC and MCM signal in the WT strain. However, these sites also harbor only weak MCM enrichment in the mutant strain (Fig. 5biii), consistent with an inefficient recruitment of MCM to these sites. Taken together, our data demonstrates that the ORC-IHΔ shows low ACS binding specificity on chromatin, and the novel non-ARS binding of the ORC-IHΔ is insufficient to promote efficient MCM loading at all sites. However, its promiscuity makes up for its inefficiency to

allow sufficient successful MCM loading to support a robust albeit slightly defective S-phase (Fig. 2c, d).

To functionally test whether the ORC-binding sites in the WT and mutant are used as replication initiation sites, we conducted bromodeoxyuridine immunoprecipitation-sequencing (BrdU-IP-seq) with cells arrested at the early S-phase by HU (Fig. 5a). While BrdU incorporation peaks align well with the ORC and MCM ChIP-seq peaks at early replication origins in the WT, consistent with usage in replication initiation, no clear peaks were observed in Orc4-IHΔ (Fig. 5a, b and Supplementary Fig. 8). The diffused signal suggests that the ORC-IHΔ-binding sites were

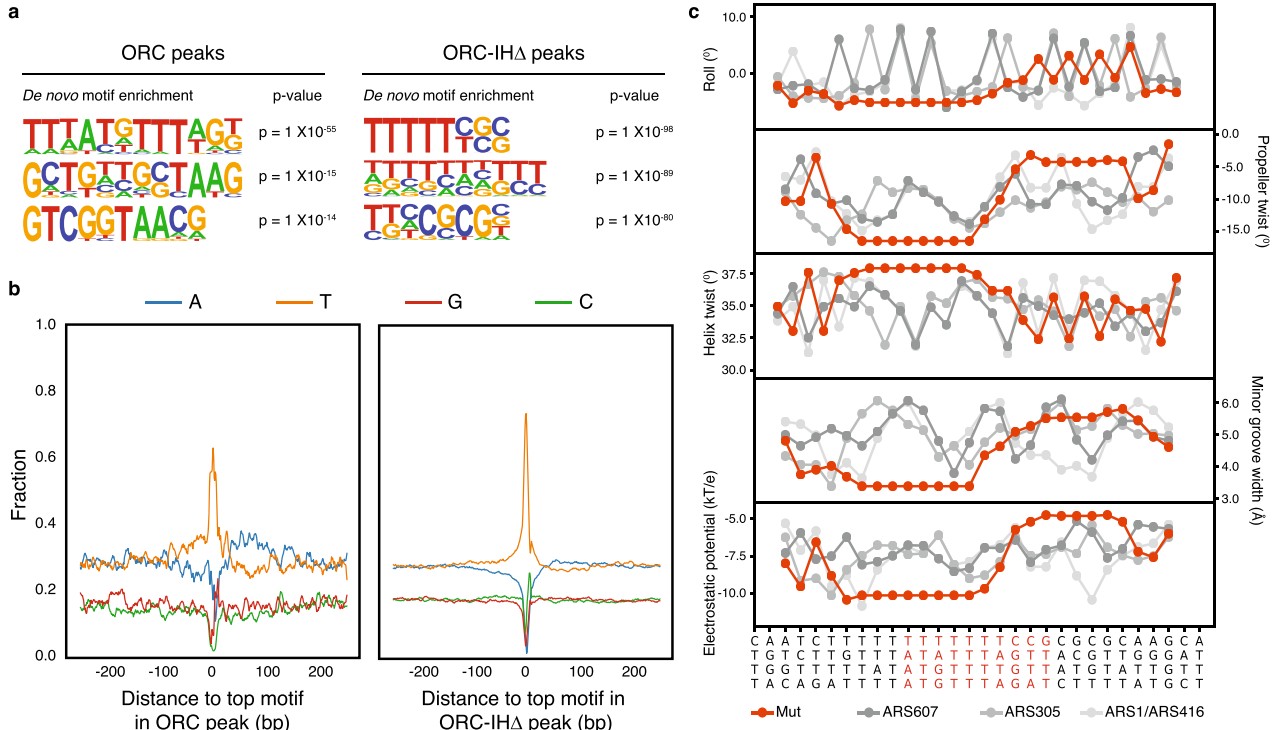

**Fig. 4 WT and mutant ORC possess differential binding affinities to distinct DNA sequences. a** De novo motif analyses reveal distinct sequences enriched in WT ORC (left) and ORC-IHΔ (right) peaks. The top three most significantly enriched sequences are shown. *P* values are calculated by binomial test using randomly selected genomic regions as background. **b** Base composition analysis of top motif within WT and Orc4-IHΔ ORC ChIP-seq peaks shows the fraction of each base. Motifs are organized in the same orientation and position 0 denotes the first base of the particular sequence. A significant enrichment of poly-T and poly-A was detected upstream and downstream of the motifs, respectively. **c** High-throughput prediction of different structural shape parameters of DNA between TTTTTTTCCGCG motif and three known ACSs from ARS607, ARS305, and ARS1. Shape features include minor groove width (MGW), propeller twist (ProT), helix twist (HelT), electrostatic potential (kT/e), and Roll. HelT and Roll describe rotation angles between adjacent base pairs. ProT describes a rotation angle between two bases that form a base pair.

stochastically activated as origins in the cell population. This disorganized firing of origins may result in delayed replication or resolution of merging forks in parts of the genome activating the DNA replication checkpoint in late S (Fig. 2d). The efficient S-phase progression of the mutant cells (Fig. 2c) corroborates with the activation of a sizable number of the ORC-IHΔ-binding sites in the cell population to give rise to the relatively high, but flat BrdU-IP-seq signals across the genome (Supplementary Fig. 8).

**ORC-IHΔ binds open chromatin sites that do not require nucleosome repositioning.** Previous work has shown that ORC exclusively binds to NDRs and in turn modulates chromatin states, as measured by micrococcal nuclease digestion-sequencing (MNase-seq)[30]. Furthermore, nucleosome positions can exert a positive or negative effect on replication initiation[31], and ORC plays a direct role in positioning nucleosomes[32,33]. Given that ORC-IHΔ has expanded binding specificity, it is important to address how it targets specific genomic loci and its potential effects on local chromatin structures. To determine whether chromatin accessibility and nucleosome positioning would play a role in influencing ORC-IHΔ binding, we conducted assay for transposase-accessible chromatin-sequencing (ATAC-seq) and MNase-seq in WT and mutant strains. Comparison of ATAC-seq peaks revealed that the overall chromatin states of WT and mutant are highly consistent (Fig. 6a, b). However, while ORC in WT has significantly higher enrichment at ARS-containing ATAC-seq peaks, ORC-IHΔ shows enrichment at open chromatin regions with no discernable preference for ARSs (Fig. 6c, Supplementary Fig. 3e and Supplementary Fig. 9).

We further investigated the nucleosome positioning at annotated ARS regions ($n = 249$) and unique ORC-IHΔ-binding sites ($n = 1,905$). ORC binding at ARSs has been shown to involve chromatin remodeling that widens the nucleosome spacing upon MCM loading[13,30]. When comparing the mean nucleosome positioning index of WT and mutant at ARS regions, we discovered a significant difference of 26 bp in the average inter-nucleosome distance at these loci (WT = 247 bp, ORC-IHΔ = 221 bp; Fig. 6d) consistent with a previous observation about the role of ORC in nucleosome positioning at ARS regions[30]. In contrast, no significant difference is found at unique ORC-IHΔ-binding regions (WT = 297 bp, ORC-IHΔ = 295 bp), suggesting that no nucleosome repositioning occurs at these sites upon mutant ORC binding (Fig. 6d). Strikingly, these ORC-IHΔ-binding regions are wider than WT ORC-binding regions by ~50 bp even after the repositioning of nucleosomes by ORC. A shift in nucleosome positions can be observed in a screenshot of an ORC-binding site unique to WT, but not at sites common to both WT and mutant or unique to mutant (Fig. 6e). Given that ORC-IHΔ has little sequence specificity, its selectivity for binding appears to be dictated by a particular physical size of the NDRs in open chromatin with certain characteristics, such as those found in TSSs and TESs (Fig. 3d) distributed across the chromatin landscape.

## Discussion

Until recently, studies of replication origins in human have been limited to methodologies that examine the chromatin landscapes of where replication initiation begins[19,34–36] rather than the

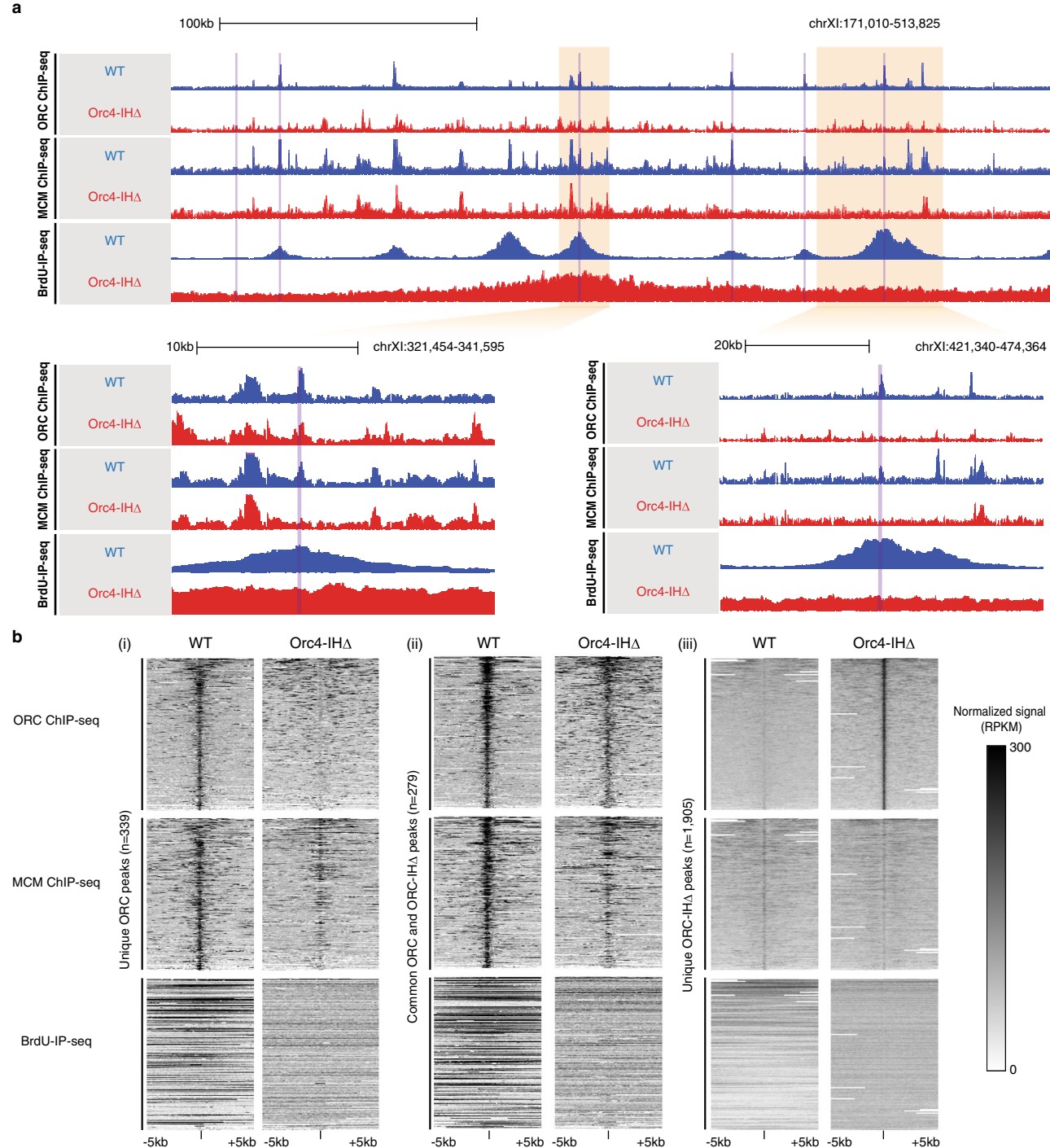

**Fig. 5 Genome-wide analysis of the function of ORC-IHΔ in MCM loading and replication initiation. a** Genome browser screenshots illustrate the changes in the ORC and MCM ChIP-seq, as well as BrdU-IP-seq normalized signals in WT and Orc4-IHΔ cells. For instance, examining a large region on chromosome XI (top), ORC, and MCM ChIP-seq enrichments show both consistency and differences between WT and ORC-IHΔ. However, BrdU incorporation appears dramatically altered. Zooming into specific loci, while some ARSs show similar binding of ORC and MCM (bottom left), others show a loss of ORC and MCM binding in Orc4-IHΔ (bottom right). This reduction is also associated with lower BrdU incorporation. Verified ARS regions (OriDB) are highlighted in purple. *Y*-axis for all ChIP-seq and BrdU-IP-seq tracks within each screenshot was adjusted to the same scale for each assay. **b** Heatmaps demonstrate the global changes of ORC and MCM ChIP-seq and BrdU-IP-seq signals. Focusing on ±5 kb surrounding ORC ChIP-seq peaks unique to WT (i), common to both WT and Orc4-IHΔ (ii), and unique to Orc4-IHΔ (iii), the normalized signals (RPKM) in WT and Orc4-IHΔ cells are shown.

precise locations of ORC-binding sites[16]. While genome-wide study of human ORC1 and ORC2 identified 50,000–100,000 binding sites across the human genome[16], no consensus sequence for ORC binding was predicted. In this study, we made use of a yeast model to examine how ORC selects its binding sites

focusing specifically on the 19aa IH of Orc4 that binds and bends the ACS[8]. Our model suggests that removal of this IH expanded ORC's DNA-binding repertoire such that it no longer preferentially binds the ACS or other predictive base sequences (Fig. 7). Instead, it acquires the propensity to bind promiscuously

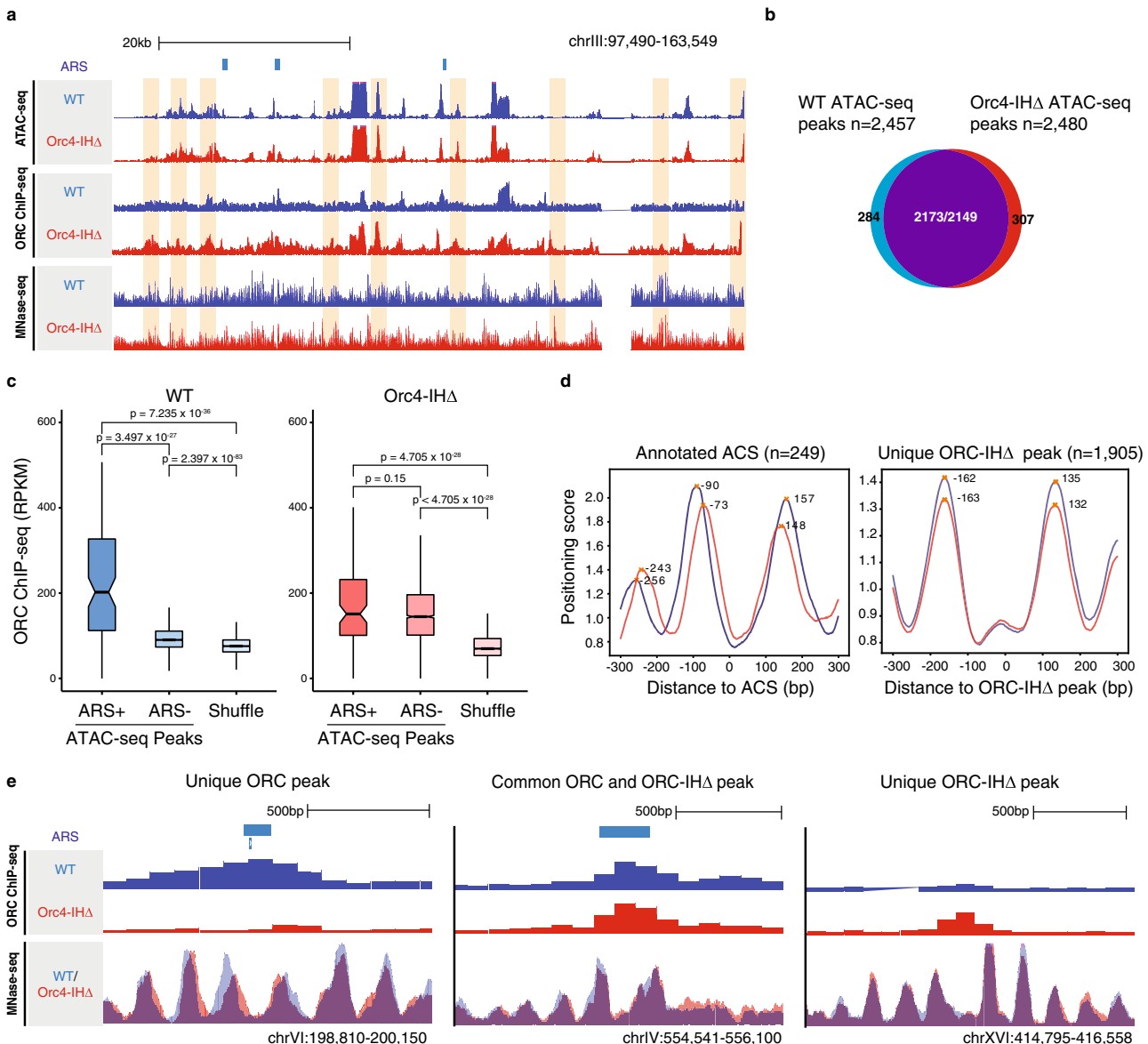

**Fig. 6 ORC-IHΔ binds promiscuously to open chromatin and fails to facilitate nucleosome repositioning at ACSs. a** A genome browser screenshot of a large region on chromosome III showing the chromatin accessibility, ORC enrichment, and nucleosome positioning in WT compared to Orc4-IHΔ. Orc4-IHΔ demonstrates a bias in binding to ATAC-seq peaks (orange shading). Annotated ARSs are denoted on the top track by light blue bars. Y-axis for all ChIP-seq, ATAC-seq, and MNase-seq tracks was adjusted to the same scale for each assay. **b** Venn diagram showing the high degree of overlap between ATAC-seq peaks defined in WT (n = 2457; blue) and Orc4-IHΔ cells (n = 2480; red). Of these, 2173 WT and 2149 Orc4-IHΔ peaks overlapped by at least 1 bp (purple), while 284 and 331 peaks were identified as unique in WT and Orc4-IHΔ respectively. **c** Boxplot showing ORC ChIP-seq enrichment (RPKM) at ATAC-seq peaks, and shuffled genomic loci in WT and Orc4-IHΔ cells. ATAC-seq peaks were further subdivided into peaks that overlap with OriDB annotated ARS (ARS+) and those that do not overlap (ARS−). In WT, pairwise comparisons show that there is significantly higher ORC binding at ARS+ peaks (n = 104) than ARS− peaks (n = 2,612) and shuffled regions (n = 2,716). However, in Orc4-IHΔ, while higher enrichment was observed as compared to the shuffled, there was no significant difference between ARS+ and ARS− peaks. The upper and the lower bounds of the boxes denote the 75th and 25th percentiles of the data, respectively. The black lines within the boxes indicate the medians. P values are calculated by two-sided Wilcoxon rank-sum test. **d** Comparison of the mean nucleosome positioning score of WT (blue) and Orc4-IHΔ (red) surrounding annotated ACS regions and unique ORC-IHΔ ChIP-seq peaks (±300 bp). Loci are centered at the start of annotated ACS regions or peak summit of unique ORC-IHΔ peaks. Crosses demarcate the local maxima. Orc4-IHΔ cells show a significant difference in the spacing between −1 and +1 nucleosomes, specifically at ACS regions compared to the WT. **e** Genome browser screenshots illustrates WT and mutant ORC-binding sites, and their corresponding nucleosome positioning. Examples are shown of ORC ChIP-seq peaks defined as unique to WT (left), common between WT and Orc4-IHΔ (middle), and unique to Orc4-IHΔ (right). Orc4-IHΔ specific difference in nucleosome positioning was detected at the unique to WT region. The ORC ChIP-seq normalized signal (RPKM) and MNase-seq normalized signal (RPKM) are shown for both WT (blue) and Orc4-IHΔ (red). Annotated ARSs are denoted on the top track by light blue bars. Y-axis of ORC ChIP-seq and MNase-seq tracks were set to the same scale for each assay.

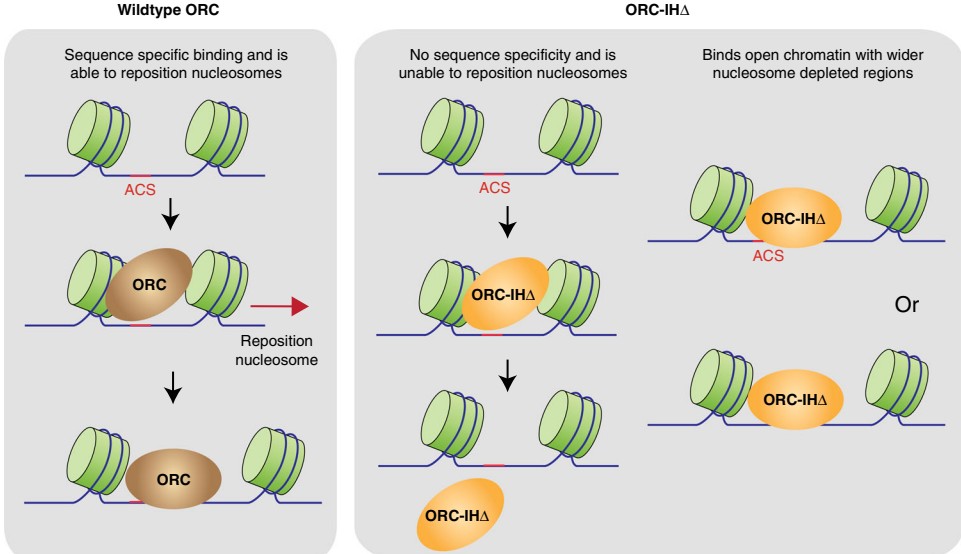

**Fig. 7 Model for the selection of binding sites by the WT and mutant ORC.** WT ORC binds ACS and is able to reposition the flanking nucleosomes upon binding[13]. ORC-IHΔ binds promiscuously to T-rich sequences, including ACSs but unable to reposition nucleosomes. Therefore, it binds wider nucleosome-depleted regions that do not require the repositioning of nucleosomes.

to intrinsically shaped sequences in nucleosome-depleted open chromatin, such as those located near upstream and downstream of actively transcribing genes (Fig. 3d). Furthermore, it selects sites that do not require nucleosome repositioning for binding (Fig. 7). Perhaps the removal of the IH may have also abolished ORC's ability to reposition nucleosomes[30,32], driving it to opt for preexisting NDRs that are extra wide for ORC binding and pre-RC assembly. Indeed, TSSs are known to be rich in poly-T tracts with extra wide NDRs and well-positioned downstream nucleosomes[13]. This inability of the mutant ORC to reposition nucleosomes may account for the loss of its binding to most ACSs in vivo, but unaffected in its affinity to ACSs on naked DNA in vitro.

So, what characterizes the new preferred binding sites for the mutant ORC? It is conceivable that the preferred three-dimensional (3D) DNA shapes characterized by poly-T tracts might play the important role of precluding the binding of nucleosomes in creating extra wide nucleosome spacing[33]. The narrow minor groove width, enhanced negative ProT, and increased HelT in the region of the poly-T tract compared to the adjacent GC-rich DNA sequence shown in the predicted DNA model (Supplementary Fig. 7) all contribute to the more rigid DNA shape with an intrinsic curvature that likely disfavors nucleosome assembly. Indeed, analysis of the human genome showed that GC-rich motifs, poly-T, and poly-A tracts form rigid structures that exclude nucleosomes[37].

The newly acquired DNA-binding properties of the yeast ORC-IHΔ are strikingly similar to many of the characteristics of the human ORC. They include the stochastic clustered associations at accessible chromatin regions[38], the preference for DNA shapes, such as promoters that potentially exclude nucleosomes[3], and the agnosticism toward base-specific sequences[16]. This study provides a link between a particular structural motif in ORC and its divergent properties that have evolved to adapt to their changing demands in vastly different life cycles. Life cycles are characterized by stages of maturation defined by the expression of distinct sets of genes in an orderly timetable. The chromatin landscape changes as quiescent genes are activated and active genes become silent. The ability of ORC to find its place in response to changing chromatin landscapes at each developmental stage is crucial for the success of a life cycle. Further

understanding of the preferred DNA shapes and nucleosome positioning requirements in model systems may provide new insights for the plasticity of the human ORC in selecting replication initiation sites during programmed development[39] and disease transformation[40].

## Methods

**Strain constructions.** Yeast strain with the endogenous *ORC4 or ORC1* under the regulation of the *GALS* promoter was generated in the W303-1a background, using PCR-based approach[41]. Plasmids for ectopic expression of *orc4-IHΔ*, *orc1-BP4Δ*, *ORC4*, or *ORC1* under their native promoters were integrated into the corresponding *GALS-ORC4* or *GALS-ORC1* yeast strains. Tandem copies of MYC and FLAG tags were fused to the C terminus of *ORC6* and *MCM7* in the relevant *GALS-ORC4* strains for genomic analysis[41]. Plasmid for overexpression of FLAG-tagged mutant Orc4 was integrated into the background strain ySD-ORC (a gift from John Diffley) that overexpresses WT Orc1-6 subunits[42] for mutant ORC protein purification. For yeast strains and plasmids used in this study see Supplementary Table 1.

**Growth conditions, cell-cycle synchronization, and FACS.** WT and mutant Orc4-IHΔ yeast strains were cultured in YPG medium before shifting to YPD medium to deplete the endogenously expressed Orc4 proteins for at least 18 h at 30 °C until $OD_{600}$ of ~0.01 starter culture reached ~2. Strains were grown at 30 °C in YPD medium for all experiments. Cells were arrested at G1 with α factor (Gen-Script) at a concentration of 10 μg/ml for ~3 h and replenished every 60 min. G2/M arrests were achieved by culturing cells in medium containing 20 μg/ml benomyl (Sigma) for ~3 h. FACS analysis was performed, as previously described[43].

**In vivo and in vitro MCM loading assays.** In vivo MCM chromatin loading assay was performed, as described[44] with the following modifications. Cell samples were incubated in spheroplasting buffer containing lyticase at 37 °C for 15 min with mixing at every 3 min until $OD_{600}$ of a 1:100 dilution of cell suspension (in 0.2% Triton X-100) dropped to <10% of the starting value. Spheroplasts were pelleted at 1700 × g for 2 min at 4 °C and washed with 0.5 ml ice-cold spheroplast wash buffer (100 mM KCl, 50 mM Hepes-KOH (pH 7.5), 2.5 mM MgCl₂, and 0.6 M sorbitol) containing 1 mM PMSF and 2 μg/ml pepstatin A. The spheroplast pellets were resuspended in 400 μl of extraction buffer (EBX) and lysed by incubation on ice for 20 min with vigorous mixing every 3 min. In vitro MCM loading assay was performed as described previously[45]. Purification of WT and mutant ORC and negative staining EM were performed as described[8] with the exception that for mutant ORC, IP was performed using anti-FLAG M2 affinity gel (Sigma) followed by further purification, using ion-exchange chromatography with mono-Q column (GE Healthcare).

**Rad53 phosphorylation assay and HU sensitivity test.** WT and mutant cells were arrested at G1, as previously described and released into S-phase in YPD medium with or without 200 mM HU. Cells were collected at different time points

post release from G1 and whole-cell protein extracts were prepared from cell pellets by trichloroacetic acid precipitation. Samples were analyzed on 6% SDS–PAGE gel. For spot analysis, ~5 ml of exponentially growing WT and mutant cells (OD$_{600}$ ~0.8) were collected, washed, and resuspended in 500 μl sterile double-distilled water. Tenfold serial dilutions of each test strain were performed and 10 μl were spotted onto YPD plates containing HU.

**ORC and MCM ChIP-seq.** For ORC and MCM ChIP, WT and mutant (Orc4-IHΔ) yeast strains with MYC-tagged Orc6 and FLAG-tagged Mcm7 were arrested at G2/ M and G1 phase, respectively. Chromatin extracts were prepared and immuno-precipitated, as previously described[46] with minor modifications. About $2 \times 10^9$ cells were crosslinked by adding formaldehyde (Millipore) to a final concentration of 1% and incubated at room temperature for 25 min with gentle shaking. Cross-linking reaction was halted by adding glycine to a final concentration of 0.125 M and incubation for an additional 10 mins at room temperature. Cells were resus-pended in lysis buffer and subjected to bead-beating, using a Mini-Beadbeater (Biospec). Chromatin fraction was collected by centrifugation and sonicated to yield an average DNA size of 300 bp, using ultrasonicator (Covaris). About 400 μl of clarified sonicated chromatin was subjected to immunoprecipitation with 60 μl of protein G-Dynabeads (Invitrogen) conjugated to anti-MYC antibody (Roche, 9E10) for Orc6-MYC ChIP or anti-FLAG antibody (Sigma, M2) for Mcm7-FLAG ChIP at 4 °C overnight. Immunoprecipitated DNA was then recovered, as pre-viously described[47]. Reverse crosslinked samples were digested with RNase A (Roche) for 1 h followed by proteinase K (Roche) for 2 h at 37 °C. DNA was purified using MinElute PCR purification column according to Qiagen protocol. Validation of the ChIP was carried out by qPCR targeting positive control loci using WT DNA (see primers in Supplementary Table 2).

**BrdU-IP seq.** For BrdU incorporation, WT, and mutant (Orc4-IHΔ) yeast strains bearing BrdU-Inc cassette was generated, as previously described[48]. Early S-phase-arrested cells with BrdU-labeled genomic DNA was extracted, fragmented, and immunoprecipitated, as described[30,49]. Validation of BrdU-IP was carried out by qPCR targeting known origins, using WT DNA (see primers in Supplementary Table 2).

**Micrococcal nuclease digestion for MNase-seq.** Chromatin was digested by MNase to mostly mononucleosomes, as previously described[30] with the following modifications. Spheroplasting of yeast cells was performed with zymolyase at 25 °C for 35 min. Conditions for MNase digestion was first optimized by adjusting the amount of the MNase added to the digestion reaction. Different dilutions of 15 units/μl MNase (NEB) were prepared to determine the best digestion conditions: 1.5-ml tubes contained 0, 1, and 2 μl (1:4 dilution in water), 1 μl (1:4 dilution in water), 2 μl (1:16 dilution in water), and 1 μl (1:16 dilution in water) of MNase, to which 400 μl of the resuspended cells was added.

**Sequencing library preparation.** Sequencing libraries for ChIP-seq, BrdU-IP-seq, and MNase-seq were prepared using KAPA HyperPrep Kit according to manu-facturer's protocol. To account for the smaller amounts of immunoprecipitated materials in ORC and MCM ChIP, adaptor ligation was performed at 4 °C over-night. Quantification of adaptor-ligated DNA was performed using quantitative real-time PCR (KAPA Library Quantification Kit) before library amplification for determination of optimal PCR cycle number. Fragment size of amplified libraries were analyzed on fragment analyzer (Agilent) and size selected with AMPure beads if needed before library pooling and sequencing.

**EMSA.** Two 30 bp long dsDNA substrates, chosen from WT ORC and ORC-IHΔ ChIP-seq peaks, were annealed from single-stranded oligonucleotides with com-plementary sequences: ACS_Fw: 5′-GCTTTGTCTTGTTTATATTTAGTTACG TTG-3′ and ACS_Re: 5′-CAACGTAACTAAATATAAACAAGACAAAGC-3′ and Mutant_motif_Fw: 5′-AATCTTTTTTTTTTTTTCCGCGCGCAAGCAC-3′ and Mutant_motif_Re: 5′-GTGCTTGCGCGCGGAAAAAAAAAAAAGATT-3′. Dif-ferent protein concentrations of WT and mutant ORC proteins were incubated with substrate DNA for 20 min at 13 °C in 13 μl of binding buffer (25 mM Hepes-KOH (pH 7.5), 100 mM KGlu, 5 mM MgAc, 5 mM CaCl$_2$, 5% glycerol, and 1 mM ATP). DNA–protein complexes were separated by 0.5% agarose gel electrophoresis stained with Sybr Safe (Invitrogen) for image detection, using Gel Documentation System (Biorad).

**ATAC-seq.** ATAC-seq was carried out with $5 \times 10^6$ G1 or G2M arrested cells[50]. In brief, WT and mutant yeast cell pellets were incubated on ice for 3 min with 50 μl of cold ATAC-RSB (10 mM Tris HCl (pH 7.4), 10 mM NaCl, 3 mM MgCl$_2$, 1× ROCHE Complete Protease Inhibitor Cocktail, 0.1% IGEPAL CA-630, 0.1% Tween-20, and 0.01% digitonin). Subsequently, 1 ml of cold ATAC-RSB with 0.1% Tween-20 (10 mM Tris HCl (pH 7.4), 10 mM NaCl, 3 mM MgCl$_2$, 1× ROCHE Complete Protease Inhibitor Cocktail, and 0.1% Tween-20) was added to the mixture. Nuclei were recovered by centrifugation and resuspended in transposase mix for tagmentation (1× TTBL and 2.5 μl TTE Mix V50 (Vazyme TD501 True-Prep DNA Library Prep Kit V2 for Illumina®), 0.01% digitonin, 0.1% Tween-20,

and PBS). The mixture was incubated at 37 °C for 15 min in a Thermomixer (1000 r.p.m.). DNA was purified using the Qiagen MinElute PCR Purification Kit. ATAC-seq libraries were amplified with the TruePrep Index Kit V2 for Illumina (Vazyme) and KAPA HiFi HotStart ReadyMix (Roche). Purified libraries were size selected with AMPure XP beads for 150–800 bp fragments, according to manu-facturer's protocol. Amplified libraries were analyzed on fragment analyzer and subsequently quantified and pooled for sequencing.

**Prediction of DNA shape and reconstruction of atomic-resolution structure.** DNA shape features were predicted using a high-throughput method based on a pentamer query table derived from Monte Carlo simulations[27–29]. Shape features include MGW, Roll, HelT, and ProT. The DNAshape method predicts a set of additional helical parameters, representing rigid-body translations and rotations between or within base pairs[27]. These helical parameters are used to place bases as building blocks into the template of a standard B-form DNA, using X3DNA[51] to generate a 3D DNA structure that assumes its sequence-dependent conformation. Rebuilt DNA structures were aligned using PyMOL along their ACS binding sites to illustrate the structural differences at their flanking regions. Electrostatic potential was calculated using a high-throughput method for data mining Poisson–Boltzmann calculations at physiologic ionic strength[28].

**Bioinformatic analysis.** Details on bioinformatics analysis are included in Sup-plementary Information.

**Reporting summary.** Further information on research design is available in the Nature Research Reporting Summary linked to this article.

## Data availability
All raw and processed sequencing data were deposited to GEO under the accession number GSE149163. Previously published datasets of WT yeast RNA-seq, WT ORC ChIP-seq, and WT MCM2-7 ChIP-seq were obtained from SRR363968, GSE16926, and GSE38032, respectively. All data are available from the authors upon reasonable request. Source data are provided with this paper.

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

## Acknowledgements
This work was supported by NSFC/RGC Joint Research Scheme (N_HKUST614/17 to B.-K.T and Y.L.Z.), the Research Grants Council (RGC) of Hong Kong (GRF16143016, GRF16104617, and GRF16103918 to Y.L.Z. and B.-K.T.; and GRF17112119 to Y.L.Z.), the Hong Kong Epigenomics Project (Lo Ka Chung Charitable Foundation to D.L.), and the National Institutes of Health (R35GM130376 to R.R.).

## Author contributions
B.-K.T. and Y.L.Z. conceived the study; C.S.K.L. performed genetics and genomics experiments; M.F.C., D.L., and V.H. carried out bioinformatic analyses; Y.Q.Z. and C.S.K.L. purified ORC and performed biochemical assays; W.H.L. performed multiple sequence alignment and helped with EMSA; J.L. and R.R. carried out computational simulations; and C.S.K.L., M.F.C., B.K.T., D.L., and Y.L.Z. prepared the manuscript.

## Competing interests
The authors declare no competing interests.
