## [Peer Review File · Nature Communications]

REVIEWER COMMENTS

Reviewer #1 (Remarks to the Author):

DNA replication must initiate from hundreds to tens of thousands of sites along eukaryotic chromosomes. Despite conservation of the origin recognition complex (ORC), origins in *S. cerevisiae* are defined by a conserved consensus sequence, the ACS, whereas metazoan origins have no or limited sequence specificity. The lack of sequence specificity in higher eukaryotes is thought to contribute to the plasticity of origin selection in different cell types and developmental stages. Recent atomic resolution structures of ORC have begun to provide insight into how ORC interacts with DNA and a potential mechanism to account for sequence specificity in yeast ORC. Specifically, the authors identify an ORC4 inter-helix domain of 19 amino acids that is specific to *S. cerevisiae* ORC and test the hypothesis that deletion of this domain would 'humanize' yeast ORC and confer non-sequence specific binding. This is an exciting study and while much of the data is consistent with altered ORC binding profiles and a perturbed DNA replication program there are several components that were less clear and need to be addressed.

Where does *orc4-ih* bind relative to TSSs? The authors report that *orc4-ih* mutants bind closer to TSSs than wildtype ORC. They should generate an aggregate plot of *orc4-ih* enrichment relative to the TSSs (oriented by direction of transcription). Presumably, *orc4-ih* mutants localize upstream of the TSS in the NDR. Is there a relationship between transcriptional strength and *orc4-ih* binding? A heatmap of ORC4-*ih* ordered by promoter strength (gene expression) would address this.

The BrdU-seq patterns in the *orc4-ih* mutants were striking compared to WT. It would be nice to see several entire chromosomes to assess how many peaks were present and whether they were associated with early replicating features (eg. centromeres). Is the intra-S-phase checkpoint intact in the *orc4-ih* mutant? If it was somehow abrogated it might account for the observed seemingly widespread BrdU accumulation.

The sequence analysis around wt and *orc4-ih* mutant peaks fails to take into account the orientation of the motifs. For example, Brierer et al., 2004 oriented the sequence composition around the T-rich strand of the ACS to reveal a T-rich ACS and downstream A-rich region (ACS matches on the opposite strand were flipped). Similarly the authors should add a sequence analysis oriented on the strand specific identified motifs for WT and *orc4-ih* mutants.

The mcm chip-seq signal in WT appears to be more closely correlated with the *orc4-ih* signal than it does with WT ORC chip-seq signal. What is the correlation between mcm-chip signal from WT and *orc4-ih* mutants with ORC chip signal from WT ORC and *orc4-ih* mutants? The concern is that the efficiency of the ChIPs are not great and that they are picking up open chromatin and highly expressed genes (Teytelman, et al., 2013).

Reviewer #2 (Remarks to the Author):

The replication origin recognition complex ORC binds DNA with very different sequence specificities in *S. cerevisiae*, *S. pombe* and metazoans. In this interesting article, the authors predict, based on a recent crystal structure of *S. cerevisiae* ORC bound to a replication origin (ARS 305) DNA fragment, that the only relevant motif for base recognition is a 19-amino acid motif, called the Orc4-Insertion helix (IH), inserting in the major groove of the ARS consensus sequence (ACS). They also predict that removal of the Orc4-IH would confer metazoan-like binding properties to the yeast ORC.

They construct such a mutant and show that it executes S phase with normal kinetics but grows

slower than WT due to a prolonged transit through G2 phase. Global chromatin retention of ORC is not affected, but the genomic distribution of ORC binding sites, as seen by ChIP-seq, is clearly different, with reduced or abolished binding at WT sites and newly detectable binding at many other sites. Sequence analysis and DNA binding motif searches at binding peaks reveal the presence of the canonical ACS motif in WT but not in the mutant. In vitro binding assays with purified WT and mutant ORC proteins show that the affinity of the mutant ORC for ACS is not reduced compared to WT, but is increased at one motif extracted from in vivo binding data. Monte Carlo modelling of DNA structure suggests that this motif may adopt a distinct "shape" from the ACSs from three different ARSs.

WT ORC is known to bind nucleosome depleted ARSs and to further enlarge nucleosome spacing upon binding. ATAC-seq and MNase-seq experiments show that chromatin states of the WT and mutant are highly similar, that ORC in WT is enriched at ARS-containing ATAC-seq peaks, but that ORC in the mutant is enriched at ATAC-seq peaks with no discernible preference for ARSs. Nucleosome spacing is reduced in the mutant at ARS regions, but is particularly wide in both strains at unique mutant-ORC binding regions. This suggests that the mutant ORC, unable to reposition nucleosomes at ARS regions (or anywhere else), favors distributed binding at particularly large, constitutive nucleosome-free gaps. Although the authors do not explicitly discuss this, this result also helps to rationalize why binding of mutant ORC at WT sites is reduced or abolished in vivo but unaffected in experiments using purified proteins and DNA.

A central role of ORC is to load the MCM double hexamer onto chromatin during the G1 phase of the cell cycle. Experiments with purified proteins show that this function is intact in the mutant ORC. ChIP-seq experiments confirm that in the mutant, MCM loading is abolished at ORC peaks that are unique to WT, is maintained - albeit attenuated - at ORC peaks common to WT and mutant, and weak at ORC peaks unique to the mutant. To functionally assess replication initiation from the ORC/MCM peaks, the authors quantify BrdU incorporation along the genome following release of G1-arrested cells into S phase in the presence of hydroxyurea. While clear peaks of BrdU are observed as expected at ORC-MCM peaks in WT, no clear peaks but instead a fairly homogeneous BrdU incorporation is observed all along the genome in the mutant. The authors suggest that the promiscuity of mutant ORC binding makes up for its inefficient MCM loading, and ensures efficient S phase progression by stochastic origin activation.

Overall these results support the claim made in the title and in the abstract that the deletion of the Orc4-IH transforms the yeast ORC, which selects sequence-specific origins, into a humanized ORC, whose selectivity is more plastic and dictated by chromatin.

I was convinced by the quality of the data and by most of the interpretations. However I think the following points should be addressed.

1. The authors claim that mutant ORC binds significantly closer to annotated TSSs than WT ORC. However there is a bias in their analysis (Fig 1e). They measured the absolute distance from each WT (n=618) or mutant (n=2,219) ORC ChIP-seq peak to their nearest TSS. Since there are many more peaks from the mutant it is not surprising to find a shorter average distance, which does not mean that ORC binding is more clustered around TSS in the mutant than in WT. The authors should compare the distance computed from the 618 WT peaks with the same distance computed from multiple samples of 618 peaks randomly chosen from the 2,219 mutant peaks.

2. The sentence " The large number of ORC-IHD binding sites compared to the limited ORC

molecules per cell (24) suggests that ORC-IHD binding in the mutant strain most likely occurs stochastically" is not satisfying to me. All biochemical reactions are stochastic, in the sense that they obey the laws of random molecular collisions. The WT ORC also binds stochastically to sites of variable affinity. The reason why WT ORC peaks are not observed at novel mutant ORC peaks is probably that ORC is limiting and therefore increased binding at ARSs means decreased binding elsewhere. Conversely, the lowered mutant ORC affinity for ARS means that a larger fraction of ORC molecules are available for binding at dispersed nucleosome-free regions.

For similar reasons I suggest changing the sentence (p.4, top paragraph) " However, these sites also harbor only weak MCM enrichment in the mutant strain (Fig. 3b,iii), consistent with a stochastic recruitment of MCM to these sites." "Stochastic recruitment" should be changed to inefficient recruitment.

3. In commenting Fig 2a, the authors state that "Base composition analysis of ChIP-seq peaks reveals high prevalence of AT-rich sequences in both strains." What I see from this Figure is that the AT content is around 60% at these sites. But is this significantly higher than the mean AT-content of the yeast genome? To my knowledge the median AT-content is 61.7% .

They also state " a clear enrichment of asymmetric poly-T- and poly-A-rich sequences immediately adjacent to the summit of ORC-IHD peaks". This is not obvious to me because the binding sites are not oriented and the graph is therefore an average of two possible random orientations.

4. In Suppl Fig 5 they compare ORC binding to either an ACS-containing 30-mer oligo or a 30-mer containing the TTTTTTTTCGCG sequence, which is one of the motifs significantly enriched at mutant ORC ChIP-seq peaks. The significance of this experiment is limited as only one novel motif was studied, and this one is not among the 3 most significant motifs shown on Fig 2b. Why did they choose this particular sequence ? Can they confirm their results with another oligo pair ?

They conclude from this experiment and Monte-Carlo simulations of the DNA structure that this motif has a different "shape" from ACS. It is not very clear to me what shape feature Fig 2d is supposed to highlight. Is it just more curved ? If so can this be quantified better ? Can we really conclude from this single example that other mutant ORC binding sites are enriched with DNA of curved or otherwise altered shapes ?

5. Mutant ORC peaks harbor only weak enrichment in the mutant strain (Fig 3b,iii), although origin licensing in vitro was unaffected (Suppl Fig 2). How do they explain this discrepancy ? Could the lack of nucleosome repositioning activity of the mutant ORC disfavor MCM loading at these sites, in addition to disfavoring ORC binding at ACS specifically in vivo (Fig 1d) but not in vitro (Suppl Fig 5)? (See the remark in my general comments " Although the authors do not explicitly discuss")

6. Perhaps the most surprising result of this study is the flat BrdU-IP-seq profile observed in the mutant, compared to localized incorporation at ORC/MCM peaks in the WT. The mutant ORC/MCM profile is certainly not as flat as the BrdU-IP-seq profile, and shows many sharp MCM peaks, so I would still have expected detectably preferential incorporation at MCM peaks. One way to get further insight would be to plot the correlation between MCM and BrdU signal in both strains. Another way would be to perform a time-course analysis of BrdU incorporation in cells synchronously released into S phase in absence of hydroxyurea, to avoid possible unexpected

effects of checkpoint activity, and to more precisely monitor BrdU incorporation at MCM peaks and elsewhere.

7. Can they propose an explanation for the extended G2 phase observed in the mutant ?

Reviewer #3 (Remarks to the Author):

The manuscript by Lee and coworkers studies the role of the "insertion helix" (IH) in ORC4 of *S. cerevisiae*. The authors convincingly show that deletion of this motif alters origin firing. Whereas the wild-type strain predominantly utilizes origin sequences that contain the ARS consensus sequence, the deletion mutant exhibits a bias toward sequences that contain an extended poly-T tract followed by a C- or G-nucleosides. The authors utilize complementary *in vivo* and *in vitro* approaches to characterize ORC1-6:DNA binding (ORC ChIP and EMSA, respectively). Moreover, they analyze pre-RC assembly (MCM ChIP) and origin firing (BrdU incorporation). Although the differences in origin activation between the mutant and wild-type strains are not quite as dramatic as described by the authors, I agree with the conclusion that "novel" peaks emerge in the mutant that are not detected in wild-type, arguing that the insertion helix of ORC4 has a role in origin specification. However, what I find puzzling, and inconsistent with the claim that origin selection is dictated by the chromatin landscape (as stated in the abstract) is the fact that the MCM binding pattern of the mutant looks almost identical to that of the wild-type strain (Fig. 3a). I think that the authors need to address this finding more critically. What these results argue is that although there are novel ORC binding sites, the pattern of pre-RC assembly remains fairly unaltered, at least for the examples displayed in Fig. 3a. I'm therefore not convinced that the ORC mutant directs pre-RC assembly based on altered chromatin configuration.

The second part of the study focuses on chromatin context of origin specification. The examples chosen for Fig. 4a suggest a very similar ATAC pattern for wild-type and mutant strains. The ORC binding pattern in this particular example is also quite similar, and so is the MNase pattern. I find this second part of the study difficult to evaluate. Fig. 4e is poorly annotated. Here it seems that certain regions of the genome show changes in the chromatin pattern, but it's almost impossible to interpret how these changes correlate with ORC binding. A more in-depth analysis of a novel ORC site using DNase I and MNase foot printing would help to better understand whether the authors' model holds true or not.

In summary, I find this work innovative and interesting. The idea of deleting the insertion helix in budding yeast ORC4 to mimic human ORC is compelling. The *in vivo* and *in vitro* ORC binding data are consistent with the claims by the authors. The mutant ORC4 strain shows a much more dispersed binding pattern with thousands of small peaks that appear to contribute to larger initiation zones reminiscent of origin activation in human cells. What I don't understand is the argument for chromatin playing a major role in determining the mutant ORC binding sites. That part of the study is not well explained to the reader. In the text, the authors refer to specific numbers that I could not trace back to the figures that were given for reference. It is therefore difficult to evaluate the second part of the manuscript.

Specific concerns:

1) Fig. 1. The Eaton reference should be included in the figure legend. Please explain how Eaton et al. relates to the more commonly used ori database (oriDB).

2) Fig. 1a) It is difficult to distinguish the structures of Sc, Dm and Hs ORC4 in the right panel. I think individual depictions might be more helpful to the reader. It remains unclear what interface is used in Dm and Hs ORC4 to contact DNA.

3) Fig 1c) The Venn diagram depicts two numbers in the common space and explains that the overlap between these sequences is by at least 1 bp. Why was that cutoff chosen? It doesn't make intuitive sense to me. How were the peaks in the ORC4 mutant defined? The majority of peaks are tiny signals. Please indicate the cutoff that was chosen over background to score a signal as peak.

4) Fig. 1e) What do you mean by "number-matched shuffled"? Please explain.

5) Fig. 3a) Why is the MCM binding pattern between wild-type and the ORC4 mutant almost identical, whereas the origin activation pattern (BrdU) looks quite different. How can this be reconciled with the differences in ORC occupancy?

5) Fig. 3b) Why is the MCM binding in the ORC4 mutant more dispersed than in wild-type cells at common peaks (ii)?

6) Fig. 4c and d) Please explain the take-aways from these data more thoroughly. How do you define WT shuffle and MT shuffle? In Fig. 4d, how do you define "region start"? Where does the number of 26bp come from? It is described on page 4 in the last paragraph, and there is a reference to Fig. 4d. Fig. 4d lacks any quantification. I have no idea where the numbers in the text (WT=247 bp, ORC-IH=221bp) come from. Please explain this better. The same is true for the other numbers mentioned in this paragraph with reference to Fig. 4d.

7) Fig. 4e) please define the color-coding. Please quantify the differences in nucleosomal spacing. Are they meaningful? Are they statistically significant?

8) Supplementary Fig. 4) Define "density". Density of what? How do you normalize for the different peak numbers in the two strains?

9) Supplementary Fig. 6). See above point 8.

10) Why were ORC peaks determined in G2/M phase (page 2 last paragraph), and not in G1?

11) It might be helpful to include a cartoon that explains the conclusions in the last part of the results that starts with "Strikingly, these ORC....." I have no idea where the numbers ("40-60 bp wider") are coming from. The reference to Fig. 4e is not helpful, because there is no quantification in Figure 4e. Please include the quantification, otherwise it is impossible to evaluate the data and the model.

We are grateful to all three reviewers for their constructive comments which we have taken seriously and addressed one by one as listed below. As a result, the paper is much improved in clarity and quality.

Note: References to new figures are highlighted in blue.

Reviewer #1

1) *“Where does *orc4-ih* bind relative to TSSs? The authors report that *orc4-ih* mutants bind closer to TSSs than wildtype ORC. They should generate an aggregate plot of *orc4-ih* enrichment relative to the TSSs (oriented by direction of transcription) . Presumably, *orc4-ih* mutants localize upstream of the TSS in the NDR. Is there a relationship between transcriptional strength and *orc4-ih* binding? A heatmap of ORC4-*ih* ordered by promoter strength (gene expression) would address this.”*

We thank the reviewer for this excellent suggestion. To address the binding of ORC-IH relative to genes and their expression levels, we integrated a published RNA-seq dataset (SRR363968) from the same WT yeast strain (W303-1A). We investigated the normalized expression levels (TPM) of all annotated genes and their relative WT and *Orc4-IHΔ* enrichment. We indeed found a positive relationship between ORC-IH binding, both upstream and downstream, and the transcriptional strength of the associated genes. This new analysis is now included in the new Fig. 1e. Because of the significance of the similarity between the mutant ORC and human ORC in their enriched binding near TSSs , we have added a sentence in the abstract to highlight this observation.

“Notably, the altered yeast ORC has acquired an affinity for regions near transcriptional start sites (TSSs) also favored by the human ORC.”

2) *“The BrdU-seq patterns in the *orc4-ih* mutants were striking compared to WT. It would be nice to see several entire chromosomes to assess how many peaks were present and whether they were associated with early replicating features (eg. centromeres). Is the intra-S-phase checkpoint intact in the *orc4-ih* mutant? If it was somehow abrogated it might account for the observed seemingly widespread BrdU accumulation.”*

We have now included entire chromosome screenshots of the BrdU-seq normalized signal of wild type and mutant in new Suppl Fig. 8. Centromeres are denoted by dotted lines. The intra-S-phase checkpoint in the *Orc4-IH* mutant is activated even under normal condition in the absence of HU (Suppl Fig. 1d), thus explains the G2-M delay and the observed seemingly widespread BrdU accumulation. We also attribute the flat BrdU-IP-seq profile in the mutant to the stochastic and inefficient firing of a small number of origins among a large number of potential initiation sites in the cell population. This effect is complicated further by the activation of the intra-S phase checkpoint that may further spread out the already sparsely distributed initiation events across the genome.

3) *“The sequence analysis around wt and *orc4-ih* mutant peaks fails to take into account the orientation of the motifs. For example, Brierer et al., 2004 oriented the sequence composition around the T-rich strand of the ACS to reveal a T-rich ACS and*

downstream A-rich region (ACS matches on the opposite strand were flipped). Similarly the authors should add a sequence analysis oriented on the strand specific identified motifs for WT and orc4-ih mutants.”

We appreciate the reviewer's comment and suggestion. We have now conducted additional analyses to address the sequence composition adjacent the motifs. We used the results of our de novo motif analysis and identified the WT and ORC-IH Δ ChIP-seq peaks that harbour these motifs. We identified 183 regions within WT ORC peaks that contain the ACS-like (TTTATGTTAGK) motif and 2,555 regions within ORC-IH Δ peaks with the novel (TTTTTYSS) motif. Orientating the motifs based on the strand, we investigated the surrounding regions' base composition with the position 0 denoting the first base of the motif. As expected from the findings from Brierer et al. 2004, we found an A-rich region downstream of the WT motif (Fig 2b). The same bias is observed in the mutant, albeit at a much lesser degree (Suppl Fig 5). This result is consistent with our findings that the ORC-IH Δ shares some common sites with WT ORC but also binds promiscuously to open chromatin. We have included the new base composition plots as well as the significance of A-T enrichment in the corresponding text (page 3).

4) The mcm chip-seq signal in WT appears to be more closely correlated with the orc4-ih signal than it does with WT ORC chip-seq signal. What is the correlation between mcm-chip signal from WT and orc4-ih mutants with ORC chip signal from WT ORC and orc4-ih mutants? The concern is that the efficiency of the ChIPs are not great and that they are picking up open chromatin and highly expressed genes (Teytelman, et al., 2013).

We thank the reviewer for his/her comment. It is indeed true that our tagged-MCM ChIP-seq datasets contain a certain degree of noise. This is a caveat of the assay. While there are peaks that arise due to non-specific antibody binding, we are nonetheless confident about the conclusions that we draw. We have now included a comparison between the genome-wide enrichment of WT and mutant MCM and ORC ChIP-seq datasets to ensure the accuracy of our findings. Firstly, we integrated a previously published WT MCM ChIP-seq dataset (Eaton et al 2010) for a meta-analysis. At MCM peaks called from this dataset, we detected similarly high enrichment in our WT dataset. Whereas MCM ChIP-seq in Orc4-IH Δ show significantly reduced binding (Suppl Fig. 2e) . No enrichment was observed at randomly selected control regions, meaning that our observation did not result from noisy data. When only considering ARS, we detected concordantly high enrichment of ORC and MCM in WT strain, while in Orc4-ih Δ mutant strain, little ORC and no MCM binding was found (Suppl Fig 3e). It is worth noting that since both WT and Orc4-IH MCM ChIPs were carried out together under the same conditions, there should be no difference in the noise in either dataset. Furthermore, to address the reviewer's concern, we analysed MCM enrichment at all ATAC-seq peaks and found no substantial enrichment in the vast majority of sites in any of the data sets (Suppl Fig. 2e), confirming that we are not simply detecting open chromatin. Taken together, we believe that our data shows that the differential ORC-IH Δ binding leads to significant shift in MCM loading onto chromatin.

Reviewer #2

1)The authors claim that mutant ORC binds significantly closer to annotated TSSs than WT ORC. However there is a bias in their analysis (Fig 1e). They measured the absolute distance from each WT (n=618) or mutant (n=2,219) ORC ChIP-seq peak to

their nearest TSS. Since there are many more peaks from the mutant it is not surprising to find a shorter average distance, which does not mean that ORC binding is more clustered around TSS in the mutant than in WT. The authors should compare the distance computed from the 618 WT peaks with the same distance computed from multiple samples of 618 peaks randomly chosen from the 2,219 mutant peaks.

We thank the reviewer for this question. Instead of measuring distance, we have now replaced Fig 1e with a plot of the the normalized expression levels (TPM) of all annotated genes and their relative WT and Orc4-IHΔ enrichment (see response to Reviewer #1 Q1). This analysis shows that there is a dramatic enrichment of ORC binding in the mutant upstream of TSS which is not observed in the WT strain as reported by Eaton et al. 2010. We also observe an enrichment downstream of the TES. These new findings are amended in the abstract and text.

2) The sentence " The large number of ORC-IHD binding sites compared to the limited ORC molecules per cell (24) suggests that ORC-IHD binding in the mutant strain most likely occurs stochastically" is not satisfying to me. All biochemical reactions are stochastic, in the sense that they obey the laws of random molecular collisions. The WT ORC also binds stochastically to sites of variable affinity. The reason why WT ORC peaks are not observed at novel mutant ORC peaks is probably that ORC is limiting and therefore increased binding at ARSs means decreased binding elsewhere. Conversely, the lowered mutant ORC affinity for ARS means that a larger fraction of ORC molecules are available for binding at dispersed nucleosome-free regions.

For similar reasons I suggest changing the sentence (p.4, top paragraph) " However, these sites also harbor only weak MCM enrichment in the mutant strain (Fig. 3b,iii), consistent with a stochastic recruitment of MCM to these sites." "Stochastic recruitment" should be changed to inefficient recruitment.

The reviewer's comment is well-taken. We have amended the text accordingly.

3) In commenting Fig 2a, the authors state that "Base composition analysis of ChIP-seq peaks reveals high prevalence of AT-rich sequences in both strains." What I see from this Figure is that the AT content is around 60% at these sites. But is this significantly higher than the mean AT-content of the yeast genome? To my knowledge the median AT-content is 61.7% .

They also state " a clear enrichment of asymmetric poly-T- and poly-A-rich sequences immediately adjacent to the summit of ORC-IHD peaks". This is not obvious to me because the binding sites are not oriented and the graph is therefore an average of two possible random orientations.

We appreciate the reviewer's comment. Similar to the comment from reviewer 1, we have now included a new base composition plot of the region flanking the WT and mutant ORC peaks, which are orientated based on their motifs (Fig. 2b). We have also included a statistical test (t-test) of each base to more accurately reflect the significance of a polar poly-A and poly-T enrichment (Suppl Fig. 5). Briefly, we randomly selected the same number of genomic regions of the same size, which was repeated 300 times. This represented the random distribution of the base composition of the yeast genome. Subsequently, we apply a one-tailed t-test, to calculate the significance of A and T enrichment. The results show a clear significant

enrichment of the poly-T and poly-A rich sequences upstream and downstream of the ORC peaks, respectively.

4) In Suppl Fig 5 they compare ORC binding to either an ACS-containing 30-mer oligo or a 30-mer containing the TTTTTTTTCCGCG sequence, which is one of the motifs significantly enriched at mutant ORC ChIP-seq peaks. The significance of this experiment is limited as only one novel motif was studied, and this one is not among the 3 most significant motifs shown on Fig 2b. Why did they choose this particular sequence ? Can they confirm their results with another oligo pair ?

Basically, from Figure 2a, we selected the most significantly enriched motif TTTTTYSS from the 2219 ORC-IH Δ peaks for experimental validation (Suppl Fig. 6). We agree that we cannot conclude from a single example though this particular example is a good representative because it has one of the highest peak signals with the sequence (TTTTTTTTCCGCG) located at the peak summit.

They conclude from this experiment and Monte-Carlo simulations of the DNA structure that this motif has a different "shape" from ACS. It is not very clear to me what shape feature Fig 2d is supposed to highlight. Is it just more curved ? If so can this be quantified better ? Can we really conclude from this single example that other mutant ORC binding sites are enriched with DNA of curved or otherwise altered shapes?

We now present the model in three different views with quantified bend angles for each DNA modelled as a supplementary (Supplemental Figure 7).

5. Mutant ORC peaks harbor only weak enrichment in the mutant strain (Fig 3b,iii), although origin licensing in vitro was unaffected (Suppl Fig 2). How do they explain this discrepancy ? Could the lack of nucleosome repositioning activity of the mutant ORC disfavor MCM loading at these sites, in addition to disfavoring ORC binding at ACS specifically in vivo (Fig 1d) but not in vitro (Suppl Fig 5)? (See the remark in my general comments " Although the authors do not explicitly discuss")

Yes, we believe that the *in vivo* and *in vitro* discrepancies in ORC binding and MCM loading most likely are due to the lack of nucleosome positioning activity of the mutant ORC to allow MCM loading at most ARSs. We have discussed this point more explicitly with the aid of a cartoon in the new Figure 5.

6. Perhaps the most surprising result of this study is the flat BrdU-IP-seq profile observed in the mutant, compared to localized incorporation at ORC/MCM peaks in the WT. The mutant ORC/MCM profile is certainly not as flat as the BrdU-IP-seq profile, and shows many sharp MCM peaks, so I would still have expected detectably preferential incorporation at MCM peaks. One way to get further insight would be to plot the correlation between MCM and BrdU signal in both strains. Another way would be to perform a time-course analysis of BrdU incorporation in cells synchronously released into S phase in absence of hydroxyurea, to avoid possible unexpected effects of checkpoint activity, and to more precisely monitor BrdU incorporation at MCM peaks and elsewhere.

We attribute the flat BrdU-IP-seq profile in the mutant to the stochastic and inefficient firing of a small number of origins among a large number of potential initiation sites in the cell population. This effect is complicated further by the activation of the intra-S phase checkpoint that may further spread out the already sparsely distributed initiation events across the genome.

7. Can they propose an explanation for the extended G2 phase observed in the mutant ?

The extended G2 phase can be explained by the intra-S phase check point activation in the mutant. This information is now included in Suppl Fig. 1d and in the text.

Reviewer #3

Main concerns:

I find puzzling, and inconsistent with the claim that origin selection is dictated by the chromatin landscape (as stated in the abstract) is the fact that the MCM binding pattern of the mutant looks almost identical to that of the wild-type strain (Fig. 3a). I think that the authors need to address this finding more critically. What these results argue is that although there are novel ORC binding sites, the pattern of pre-RC assembly remains fairly unaltered, at least for the examples displayed in Fig. 3a. I'm therefore not convinced that the ORC mutant directs pre-RC assembly based on altered chromatin configuration.....Fig 4e is poorly annotated.

We apologize for the confusion that we may have generated in the presentation of our data. First, we should emphasize that there is little or no alteration in chromatin configuration between WT and mutant based on our ATAC-seq analysis (Fig 4 a, b). However, the mutant ORC has altered properties such that it no longer binds DNA with base sequence specificity. Instead, it binds open chromatin (based on ATAC-seq) where the NDR are wider than those where ARSs are located, in particular, those large NDRs typically where TSSs are located. We surmise that the TTTTTYSS motif associated with these mutant ORC binding sites assume a certain 3-dimensional DNA shape that has particularly high affinity for ORC-IHΔ. These subtleties cannot be clearly discerned just based on the screenshots shown in Fig 3a. We have also revised the figure legend for Fig. 4e for clarity.

We have expanded the argument that chromatin and certain DNA “shape” often associated with poly dT tracts and TSSs play a major role in determining the mutant ORC binding sites with the aid of a cartoon model in Fig. 5.

Specific concerns:

1) Fig. 1. The Eaton reference should be included in the figure legend. Please explain how Eaton et al. relates to the more commonly used ori database (oriDB).

We thank the reviewer for this comment. We have now included the Eaton et al. reference in the Suppl 3 figure legends and the rationale for using the Eaton data as a reference for our study in the main text “.....because of their stringent criteria for defining ACSs based on T strand polarity and precise nucleosome positioning”.

2) Fig. 1a) It is difficult to distinguish the structures of Sc, Dm and Hs ORC4 in the right panel. I think individual depictions might be more helpful to the reader. It remains unclear what interface is used in Dm and Hs ORC4 to contact DNA.

We have replaced the superimposed structures with individual depictions in Fig. 1a.

3) Fig 1c) The Venn diagram depicts two numbers in the common space and explains that the overlap between these sequences is by at least 1 bp. Why was that cutoff chosen? It doesn't make intuitive sense to me. How were the peaks in the ORC4 mutant defined? The majority of peaks are tiny signals. Please indicate the cutoff that was chosen over background to score a signal as peak.

The Venn diagram shows the overlap between any peaks called in the two datasets. It is possible for a peak in 1 dataset to overlap with more than 1 peak in the other dataset. The two numbers in the common space represent the number of peaks in either WT or mutant that overlap.

Given the resolution of ChIP-seq data, we do not expect perfect overlap of peaks defined. The ChIP'ed DNA are of heterogeneous sizes and therefore can produce offsets to peak-calls. We therefore chose a cutoff (at least 1 bp overlap) to ensure direct overlap of peaks while retaining the largest number of loci for subsequent analysis. It should be noted that we applied fairly strict peak calling thresholds (FDR=0.005) and only peaks that are commonly defined in replicates are used. Although it is possible to increase the threshold for defining overlapping peaks, taken together with the analysis of the ChIP-seq signal at these regions (Figure 1d), we feel that these thresholds are sufficiently stringent.

4) Fig. 1e) What do you mean by "number-matched shuffled"? Please explain.

Number-matched shuffled loci refer to a set of randomly picked regions in the genome that have the same number of events and share the same length as the original set. For instance, for the 618 WT Orc peaks, we randomly picked 618 regions by shuffling the genomic coordinates of the peaks, whilst keeping the peak size constant. The result would be the same number of peaks that are scattered throughout the genome but share the same lengths as the test set. We would subsequently use this number-matched and size-matched shuffled set as a representative of the genome background. We have added these details to the methods.

5) Fig. 3a) Why is the MCM binding pattern between wild-type and the ORC4 mutant almost identical, whereas the origin activation pattern (BrdU) looks quite different. How can this be reconciled with the differences in ORC occupancy?

The resolution of screenshot may have caused some confusion. In order to illustrate the scope of the phenomenon, a zoomed out screenshot is required; however, at this large scale, changes may be difficult to see. A global analysis shows that MCM and ORC binding patterns are closely associated in the WT strain but differ in the mutant (Suppl. Fig 3e). This is also more clearly demonstrated in Figure 3b.

Please also see answer to Q4 of Reviewer 1

5) Fig. 3b) Why is the MCM binding in the ORC4 mutant more dispersed than in wild-type cells at common peaks (ii)?

As explained in Q3, the common peaks are likely non-overlapping binding sites recognized by the WT or mutant ORC, respectively, that are close together. So these binding sites can be treated as two independent sets.

6) Fig. 4c and d) Please explain the take-aways from these data more thoroughly. How do you define WT shuffle and MT shuffle? In Fig. 4d, how do you define “region start”? Where does the number of 26bp come from?

It is described on page 4 in the last paragraph, and there is a reference to Fig. 4d. Fig. 4d lacks any quantification. I have no idea where the numbers in the text (WT=247 bp, ORC-IH=221bp) come from. Please explain this better. The same is true for the other numbers mentioned in this paragraph with reference to Fig. 4d.

7) Fig. 4e) please define the color-coding. Please quantify the differences in nucleosomal spacing. Are they meaningful? Are they statistically significant?

We thank the reviewer for this comment. We have now replaced the heatmaps in Fig. 4d with line plots with precise locations of the flanking nucleosomes to more clearly convey the message. Essentially, we observed that at ACS regions in the mutant, the mean ± 1 nucleosome position surrounding the ACS is closer than WT because they are unoccupied. The mean size of the nucleosome free region was 247bp and 221bp in WT and mutant, respectively. Hence, we report a widening of 26bp by ORC. This difference was not detected in the Orc4-IH Δ ChIP-seq peaks, which showed unchanged nucleosome positioning surrounding the loci. We surmise that ORC-IH Δ may have lost its ability to position nucleosomes and opt for wider sites that can accommodate its occupancy. The screenshots in Figure 4e are simply examples to illustrate the 3 categories of regions (ARS with WT ORC binding, WT and ORC-IH Δ common binding and ORC-IH Δ only binding) and their corresponding nucleosome positioning. The quantification and statistical testing are provided in Fig 4c and 4d. We have also amended the relevant legends to clarify this point.

8) Supplementary Fig. 4) Define “density”. Density of what? How do you normalize for the different peak numbers in the two strains?

9) Supplementary Fig. 6). See above point 8.

Suppl Fig 4 and Suppl Fig 6 (now Suppl Fig 9) show probability density curves of random sampling, which was done 10,000 times to pick the same number of regions of the same sizes as the actual set of peaks from the genome. Therefore, the resulting curve shows a random distribution and the density refers to the probability of finding a number of overlaps with the test set. For instance, in Suppl Fig. 4a, when we randomly select 618 regions from the genome 10,000 times, we find the chance of having 250 of the sites overlapping with intergenic regions is close to 0. Whereas the probability of having 300 site overlap with intergenic definitions is approximately 0.03. The red line denotes number of overlaps between the actual ORC peaks and intergenic regions. We then apply a Wilcoxon Signed Rank Test to measure the significance between the observed number over random. To account for the different number of peaks in the 2 strains, we match the number of locus in the random selection to the specific set

to be examined. Likewise, for Suppl Fig 4b, we randomly pick 2,219 regions 10,000 times for comparison with the 2,219 ORC-IH Δ peaks. We have amended the legends and the methods to clarify these points.

10) Why were ORC peaks determined in G2/M phase (page 2 last paragraph), and not in G1?

It has been previously shown that immunoprecipitation of ORC-origin DNA complexes is largely inhibited by the pre-RC (Aparicio et al 1997 Cell, page 3). Therefore, genome-wide analyses of ORC binding sites in yeast cells were normally performed with G2/M cells (Wyrick et al 2001 Science; Xu et al 2006 BMC Genomics; Eaton et al 2010 G&D; Belsky et al 2015 G&D). Notably, it has been shown that the binding pattern of yeast ORC remains constant throughout cell cycle (Aparicio et al 1997 Cell; Liang and Stillman 1997 G&D).

11) It might be helpful to include a cartoon that explains the conclusions in the last part of the results that starts with “Strikingly, these ORC.....” I have no idea where the numbers (“40-60 bp wider”) are coming from. The reference to Fig. 4e is not helpful, because there is no quantification in Figure 4e. Please include the quantification, otherwise it is impossible to evaluate the data and the model.

We thank the reviewer for the suggestion. We have now included a cartoon in Figure 5 to explain the conclusions in the last part of the results. Furthermore, the new Figure 4d (also shown in answer to Q6 above) provides the necessary quantitation of the nucleosome positions flanking the ORC binding sites. We have also an expanded figure legend for Fig. 4e.

REVIEWERS' COMMENTS

Reviewer #1 (Remarks to the Author):

I am satisfied with the revisions and additional data analysis.

Reviewer #2 (Remarks to the Author):

I am reasonably satisfied with the authors' answers to all of my comments. Several of my concerns were shared by the other referees. I also think the authors have reasonably well answered the additional points of the other referees.

Reviewer #3 (Remarks to the Author):

The authors have addressed all of my concerns quite thoroughly. The revised manuscript is more accessible than the original version. The inclusion of the Rad53 activation data and improved documentation strengthen the study. The inclusion of a new model figure (Figure 5) is also helpful. In my mind, the data support the major claims, and this is a very timely and interesting paper.

Minor suggestions:

Page 4, line 178, "at G2/M.....at G1." change to "in G2/M....in G1."

Page 6, line 257, should read "characterizes" not "characterize"

Anja Katrin Bielinsky

REVIEWERS' COMMENTS

Reviewer #1 (Remarks to the Author):

I am satisfied with the revisions and additional data analysis.

Reviewer #2 (Remarks to the Author):

I am reasonably satisfied with the authors' answers to all of my comments. Several of my concerns were shared by the other referees. I also think the authors have reasonably well answered the additional points of the other referees.

Reviewer #3 (Remarks to the Author):

The authors have addressed all of my concerns quite thoroughly. The revised manuscript is more accessible than the original version. The inclusion of the Rad53 activation data and improved documentation strengthen the study. The inclusion of a new model figure (Figure 5) is also helpful. In my mind, the data support the major claims, and this is a very timely and interesting paper.

Minor suggestions:

Page 4, line 178, "at G2/M.....at G1." change to "in G2/M....in G1."

done

Page 6, line 257, should read "characterizes" not "characterize"

done